

# Characterizing (non-)Markovianity through Fisher information

**Paolo Abiuso, Matteo Scandi, Dario De Santis and Jacopo Surace**

ICFO-Institut de Ciencies Fotoniques, The Barcelona Institute of Science and Technology, Castelldefels (Barcelona), 08860, Spain

## Abstract

A non-isolated physical system typically loses information to its environment, and when such loss is irreversible the evolution is said to be Markovian. Non-Markovian effects are studied by monitoring how information quantifiers, such as the distance between physical states, evolve in time. Here we show that the Fisher information metric emerges as a natural object to study in this context; we fully characterize the relation between its contractivity properties and Markovianity, both from the mathematical and operational point of view. We prove, for classical dynamics, that Markovianity is equivalent to the monotonous contraction of the Fisher metric at all points of the set of states. At the same time, operational witnesses of non-Markovianity based on the dilation of the Fisher distance cannot, in general, detect all non-Markovian evolutions, unless specific physical postprocessing is applied to the dynamics. Finally, we show for the first time that non-Markovian dilations of Fisher distance between states at any time correspond to backflow of information about the initial state of the dynamics at time 0, via Bayesian retrodiction. All the presented results can be lifted to the case of quantum dynamics by considering the standard CP-divisibility framework.

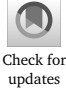

# 1   Introduction

The information contained in an evolving open system may undergo two possible dynamical regimes, dubbed Markovian or non-Markovian. Markovian evolutions are characterized by memoryless environments, where at each time the dynamical trajectory can be represented by physical transformations that solely depend on the immediately previous time step [1, 2]. It follows that in this regime the information contained in the system undergoes a monotonic degradation. On the other hand, non-Markovian dynamics are distinguished by memory effects, meaning that at any time the evolution might in general depend on all the previous time steps of the trajectory. This property allows information to flow back into the system. The complete characterization of non-Markovianity passes through the classification of the possible backflow phenomena that these evolutions can offer. This task is at the core of the non-Markovian witnessing problem [3, 4]. More specifically, one needs to find information quantifiers and specific initializations of the system (which may include ancillas) that are able to signal the non-Markovian nature of the evolution via revivals of an otherwise monotonically decaying information. Different quantifiers have been considered in this context, such as for state discrimination [5, 6], channel capacity [7], volume of accessible states [8] and correlations [9, 10]. This approach allows to understands how we can gain benefits from non-Markovian evolutions and, at the same time, it is committed to test the correspondence between its phenomenological and mathematical definition. Notice that this is relevant both for the dynamics of classical stochastic systems, as well as that of open quantum systems. In the following, we will set our discussion on the classical case, which allows us to introduce all the relevant physical and mathematical insights in a straightforward manner. However, all the presented framework and results can be extended to the case of quantum dynamics, via few tedious technical steps (cf. Appendices C and D).

A quantity that has attracted a lot of attention from the community is the trace distance. In the classical case, it is given by

$$D_{\text{Tr}}(\boldsymbol{p}, \boldsymbol{q}) = |\boldsymbol{p} - \boldsymbol{q}| = \sum_i |p_i - q_i|, \tag{1}$$

where $\boldsymbol{p}$ and $\boldsymbol{q}$ are two classical probability distributions. Such distance quantifies the distinguishability between the states $\boldsymbol{p}$ and $\boldsymbol{q}$, as the error probability of distinguishing the two in a single-shot measurement is lower bounded by $P_{\text{err}} = \frac{2 - |\boldsymbol{p} - \boldsymbol{q}|}{4}$ [11]. Moreover, this quantity decreases under physical maps, leading to a monotonic decrease for Markovian evolutions [3] (cf. Appendix A). Hence, one consequence associated to the loss of information due to Markovianity is a continuous decrease in the ability of an agent to discriminate between any two states.

Still, the effects linked to Markovianity go well beyond what can be quantified by the trace distance alone. In the following we study an alternative quantifier: the Fisher information

distance. This can be defined from its value between two infinitesimally close points. That is, given a small perturbation $|\boldsymbol{d}| \ll 1$ it reads, at leading order $\mathcal{O}(|\boldsymbol{d}|^2)$,

$$D^2_{\text{Fish}}(\boldsymbol{p}, \boldsymbol{p} + \boldsymbol{d}) \simeq \langle \boldsymbol{d}, \boldsymbol{d} \rangle_{\boldsymbol{p}} := \frac{1}{2} \sum_i \frac{d_i^2}{p_i}, \tag{2}$$

where we define the Fisher scalar product

$$\langle \boldsymbol{a}, \boldsymbol{b} \rangle_{\boldsymbol{p}} := \sum \frac{a_i b_i}{2 p_i}. \tag{3}$$

The Fisher distance between two infinitesimally close points is sometimes called *Fisher Information*. More precisely, this is defined when considering the state $\boldsymbol{p}_\theta$ to be parametrized by some variable $\theta$. Then it is possible to define

$$\lim_{\delta\theta \to 0} 2 \frac{D^2_{\text{Fish}}(\boldsymbol{p}_\theta, \boldsymbol{p}_{\theta+\delta\theta})}{\delta\theta^2} = \sum_i \frac{(\partial_\theta p_i)^2}{p_i}. \tag{4}$$

This quantity is usually termed Fisher Information (w.r.t. $\theta$), and in the rest of this work we will consider it a synonym of the infinitesimal Fisher distance, or Fisher metric (2), as they coincide up to constant factor. Moreover, while only making use of the local properties of the Fisher metric, our main results in Theorems 1,2 and 3 are statements that hold also when integrating the Fisher metric to a finite distance.[1] The Fisher Information has numerous operational interpretations: in metrology it is used to derive the Cramér-Rao bound [15,16], a fundamental limit on the precision with which $\theta$ can be estimated; it quantifies the asymptotic distinguishability between multiple copies of two states (Chernoff bound [17]); moreover, it also arises as the infinitesimal expansion of the relative entropy [18] (in fact, it can be shown that any $f$-divergence locally behaves as the Fisher information [19]).

While the relation between Fisher metric and Markovianity has been previously partially analyzed [20] here we characterize it completely. In fact, a key characterisation of the Fisher metric is given by the Chentsov's theorem: this says that the Fisher information is the unique Riemannian metric that contracts under the action of all stochastic maps [21,22] (for the quantum case, see Petz [23]). This is the starting point of our work. In particular, we study whether this strong relation between stochasticity and contractivity of the Fisher information can be reversed. That is, is it true that a map is Markovian *if and only if* it contracts monotonically the Fisher information? What are the operational consequences of the contraction/dilation of such metric?

---

[1]Notice that the metric (2) coincides with the differential of the square root of $\boldsymbol{p}$. That is, at leading order, $D^2_{\text{Fish}}(\boldsymbol{p}, \boldsymbol{p} + \boldsymbol{d}) = 2|\sqrt{\boldsymbol{p}} - \sqrt{\boldsymbol{p} + \delta\boldsymbol{p}}|^2$. This means that the Fisher metric maps the set of states to a sphere sector, via the change of variable $\mathbf{y} = \sqrt{\mathbf{p}}$ [12, 13]. Accordingly, it is possible to integrate Eq. (2) over the set of physical states, which is simply the set of positive normalized vectors $\mathcal{S}(\mathbb{R}^N) := \{\boldsymbol{p} \in \mathbb{R}^N | p_i \geq 0 \wedge \sum_i p_i = \sum_i (\sqrt{p_i})^2 = 1\}$, to obtain the finite Fisher distance [12, 14]

$$D^{\text{B}}_{\text{Fish}}(\boldsymbol{p}, \boldsymbol{q}) = \sqrt{2} \arccos(\sqrt{\boldsymbol{p}} \cdot \sqrt{\boldsymbol{q}}). \tag{5}$$

The expression just obtained corresponds to integrating on the surface of the sphere. On the other hand, one could consider the Euclidean distance between two vectors, which is given by:

$$D^{\text{H}}_{\text{Fish}}(\boldsymbol{p}, \boldsymbol{q}) = \sqrt{2|\sqrt{\boldsymbol{p}} - \sqrt{\boldsymbol{q}}|^2} = 2\sqrt{1 - \sqrt{\boldsymbol{p}} \cdot \sqrt{\boldsymbol{q}}}. \tag{6}$$

It should be noticed that the corresponding geodesic is given by the segment connecting $\boldsymbol{p}$ to $\boldsymbol{q}$. Hence, in this case the intermediate points of the optimal trajectory lay outside of the space of normalized vectors (while retaining positivity). In the literature, $D^{\text{B}}_{\text{Fish}}$ and $D^{\text{H}}_{\text{Fish}}$ are known respectively as the *Bhattacharyya distance* and the *Hellinger distance*. Here, we will not distinguish between the two, as each is a monotonous function of the other, and they locally coincide.

## 2 Framework

Before passing to the main treatment, we introduce here the main objects of interest for the rest of the work. A stochastic map (or *channel*) $T^{(t,0)}$ is a linear operator that evolves probability vectors (or *states*) $\boldsymbol{p} \in \mathcal{S}(\mathbb{R}^N)$, in the symplex defined as $\mathcal{S}(\mathbb{R}^N) := \{\boldsymbol{p} \in \mathbb{R}^N | p_i \geq 0 \wedge \sum_i p_i = 1\}$, from a time 0 to $t$,

$$T^{(t,0)}[\boldsymbol{p}(0)] = \boldsymbol{p}(t). \tag{7}$$

It follows from this definition that, for every $t$,

$$\sum_i T^{(t,0)}_{ij} = 1, \quad T^{(t,0)}_{ij} \geq 0 \ \forall i,j. \tag{8}$$

In fact, the elements $T^{(t,0)}_{ij}$ can be interpreted as the conditional probability of ending up in microstate $i$ at time $t$ starting from $j$ at time 0, i.e. $T^{(t,0)}_{ij} = P(i,t|j,0)$.

Assuming the channel to be continuous and differentiable in time, as well as invertible,[2] one can define the intermediate channel $T^{(t,s)}$ between two increasing times $s$ and $t$ as

$$T^{(t,s)} \equiv T^{(t,0)} \circ T^{(s,0)-1}, \quad t \geq s \geq 0. \tag{9}$$

Channels with these properties constitutes the class of *smooth evolutions*. From its definition, it is easy to verify that $T^{(t,s)}$ still satisfies $\sum_i T^{(t,s)}_{ij} = 1 \ \forall j$, which can be equivalently rewritten as

$$T^{(t,s)}_{jj} = 1 - \sum_{i \neq j} T^{(t,s)}_{ij}, \quad \forall j, \tag{10}$$

and corresponds to the requirement that the dynamics preserves the normalisation.

In the limit of infinitesimal time-steps, a smooth evolution $T^{(t,0)}$ is generated by the rate matrix

$$R^{(t)} \equiv \lim_{\delta t \to 0} \frac{T^{(t+\delta t, t)} - \mathbb{1}}{\delta t}, \tag{11}$$

such that $\frac{\mathrm{d}}{\mathrm{d}t} T^{(t,0)} = R^{(t)} \circ T^{(t,0)}$. Thanks to the condition in Eq. (10) (see also Eq. (14a) below), one can always decompose $R^{(t)}$ as

$$R^{(t)} = \sum_{i \neq j} a^{(t)}_{i \leftarrow j} (|i\rangle\langle j| - |j\rangle\langle j|), \tag{12}$$

where $a^{(t)}_{i \leftarrow j}$ are real coefficients called *rates*.

A smooth evolution $T$ is called *stochastic-divisible* or *Markovian*[3] [2,3] if for any partition of the interval $[0, t]$ it can be split into intermediate channels, all of which are stochastic

$$T^{(t,0)} = T^{(t,t_{k-1})} \circ T^{(t_{k-1},t_{k-2})} \circ \cdots \circ T^{(t_1,0)}. \tag{13}$$

Since the composition of two Markovian evolutions is again Markovian, one can check the stochastic-divisibility of a channel by studying maps of the form $T^{(t+\delta t,t)} \approx \mathbb{1} + \delta t \, R^{(t)}$ for

---

[2]Invertibility of the maps can can be always physical satisfied by introducing undetectable $\varepsilon$-noises to the dynamics.

[3]Markovian evolutions are an essential tool for the description of multi-time process $\{t_0, t_1, \dots\}$ where the evolution of the state only depends on the latest previous sampling of it, i.e. $P(i_k, t_k | j_{k-1}, t_{k-1}; j_{k-2}, t_{k-2}; \dots) = P(i_k, t_k | j_{k-1}, t_{k-1})$, a condition equivalent to Eq. (13) [3].

all possible times. is, we require $T^{(t+\delta t,t)}$ to be stochastic, imposing on the rate matrix, via Eq. (8),

$$\sum_i (\delta_{ij} + \delta t R^{(t)}_{i,j}) = 1 + \delta t \sum_i R^{(t)}_{i,j} = 1, \qquad \forall j, \tag{14a}$$

$$\delta_{ij} + \delta t R^{(t)}_{i,j} \geq 0, \qquad \forall\, i,j, \tag{14b}$$

where $\delta_{ij}$ denotes the Kronecker delta. The first conditions implies that $\sum_i R^{(t)}_{i,j} = 0$, which leads to the canonical decomposition (12). If we now focus on the second condition in Eq. (14) we see that $R^{(t)}_{i,j} \geq 0$ whenever $i \neq j$. In the parametrisation above this means that $a^{(t)}_{i \leftarrow j} \geq 0$. It follows that the rate matrices generating stochastic evolutions form a cone, whose elements are matrices of the form (12) having all the rates $a_{i \leftarrow j}$ positive. Since a smooth evolution $T$ is Markovian if and only if $T^{(t+\delta t,t)}$ is stochastic $\forall t$, we can give the following definition:

**Definition 1.** A smooth evolution is Markovian if and only if for all times $t$ the rates $a^{(t)}_{i \leftarrow j}$ are positive for all pairs of microstates $i \neq j$.

**Remark.** In the case of quantum dynamics the evolutions are given by Completely Positive and Trace Preserving (CPTP) superoperator $\mathcal{T}^{(t,0)}$ acting on density matrices, giving rise to the evolution $\mathcal{T}^{(t,0)}[\rho(0)] = \rho(t)$. In analogy to Eq. (13), we adopt the canonical definition of quantum Markovianity as CP-divisibility [3], i.e., $\mathcal{T}^{(t,0)}$ is said to be Markovian if for any partition of the interval $[0, t]$, it can be written as the composition of intermediate maps that are CPTP. As for classical dynamics, Markovianity of smooth quantum evolutions becomes equivalent to the positivity of the rates of the master equation [24]. For simplicity of exposition, all the following results are presented in the classical scenario, but thanks to a technical Lemma (Appendix C) we can lift them to the quantum regime. The precise proofs are deferred to Appendix D.

## 3 Contractivity of the Fisher metric and non-Markovianity detection

From now on we will omit time-dependency when no confusion can arise. The Fisher distance between any two points decreases under the action of stochastic maps. This directly implies that under Markovian dynamics the Fisher metric contracts continuously. Indeed, simple algebra (cf. Appendix B) yields, for infinitesimal $\boldsymbol{d}$,

$$\frac{\mathrm{d}}{\mathrm{d}t} D^2_{\mathrm{Fish}}(\boldsymbol{p}, \boldsymbol{p} + \boldsymbol{d}) = -\sum_{i \neq j} a_{i \leftarrow j} I_{i \leftarrow j} \leq 0, \tag{15}$$

where we implicitly defined

$$I_{i \leftarrow j} := \frac{1}{2} \left( \frac{d_i}{p_i} - \frac{d_j}{p_j} \right)^2 p_j, \tag{16}$$

as the *Fisher information flow* associated to the rate $a_{i \leftarrow j}$. These are positive objects, so that the contraction of the Fisher metric is directly associated to the positivity of the rates $a_{i \leftarrow j}$. The quantity in Eq. (15), that is, the rate of contraction of the Fisher metric is the main object of study in this work.

Note that, such derivative is physically defined at time $t$ only for points in the image of $T^{(t,0)}$, while other points of $\mathcal{S}(\mathbb{R}^N)$ are not guaranteed, in general, to be physical states after

applying the subsequent evolution $T^{(t',t)}$. However, by considering the infinitesimal evolution $T^{(t+\delta t,t)} \simeq \mathbb{1} + \delta t R^{(t)}$, for each point *in the interior* of $\mathcal{S}(\mathbb{R}^N)$ there exists a $\delta t$ small enough so that the corresponding evolved state is still physical. This allows the rate of contraction $\frac{d}{dt} D_{\text{Fish}}(\boldsymbol{p}, \boldsymbol{q})|_t$ to be well defined at all times for all points in the interior of $\mathcal{S}(\mathbb{R}^N)$. This is also the same set of points where the Fisher metric itself (2) is non-singular.

Now suppose that for some time $t$ there is a negative rate $a^{(t)}_{\tilde{i} \leftarrow \tilde{j}} < 0$ (i.e., the evolution is non-Markovian). Is this sufficient to reverse the contraction of the Fisher information? The positive answer is given by the following

**Theorem 1.** *A smooth evolution $T^{(t,0)}$ is Markovian if and only if it induces a decrease in Fisher distance between any two points in $\mathcal{S}(\mathbb{R}^N)$ at all times, i.e.*

$$T^{(t,0)} \text{ is Markovian} \iff D_{\text{Fish}}\left(T^{(t+\delta t,t)}[\boldsymbol{p}], T^{(t+\delta t,t)}[\boldsymbol{q}]\right) \leq D_{\text{Fish}}(\boldsymbol{p}, \boldsymbol{q}) \, \forall t, \, \forall \boldsymbol{p}, \boldsymbol{q} \in \mathcal{S}\left(\mathbb{R}^N\right). \quad (17)$$

*Proof.* Chentsov theorem guarantees that the Fisher distance contracts under stochastic maps. This is also confirmed by our Eq. (15). Moreover, notice that local contractivity of a metric always implies contractivity of the global distance, thanks to the triangle inequality. As the $\Rightarrow$ implication in Eq. (17) is trivial from the above discussion, only the proof of the opposite $\Leftarrow$ is needed. For that, suppose that the first instance of non-Markovianity happens between time $t$ and $t + \delta t$. In order to prove the statement it is sufficient a single counterexample, which we can find locally, considering the Fisher distance between any two infinitesimally close points $\boldsymbol{p}$ and $\boldsymbol{p} + \boldsymbol{d}$, with $|\boldsymbol{d}| \ll 1$, evolving according to $T^{(t+\delta t,t)}$. Then, for any negative rate $a^{(t)}_{\tilde{i} \leftarrow \tilde{j}} < 0$, one can find a point $\boldsymbol{p}$ and a perturbation $\boldsymbol{d}$ such that $\frac{d}{dt} D^2_{\text{Fish}}(\boldsymbol{p}, \boldsymbol{p} + \boldsymbol{d}) > 0$. In fact, assume without loss of generality that $a_{1 \leftarrow 2} < 0$ and consider $\boldsymbol{p}$ and $\boldsymbol{d}$ of the form

$$\boldsymbol{p} = \begin{pmatrix} \mathcal{O}(\varepsilon) \\ 1 - \varepsilon \\ \mathcal{O}(\varepsilon) \\ \vdots \\ \mathcal{O}(\varepsilon), \end{pmatrix}, \quad \boldsymbol{d} = \begin{pmatrix} \mathcal{O}(\varepsilon) \\ \mathcal{O}(\varepsilon) \\ \mathcal{O}(\varepsilon^2) \\ \vdots \\ \mathcal{O}(\varepsilon^2) \end{pmatrix}, \quad (18)$$

where $\varepsilon$ is an arbitrary small number and the vectors $\boldsymbol{p}$ and $\boldsymbol{p} + \boldsymbol{d}$ are properly normalised. Notice that it is always possible to choose $\boldsymbol{p}$ and $\boldsymbol{p} + \boldsymbol{d}$ in the interior of $\mathcal{S}(\mathbb{R}^N)$ (i.e., with strictly positive components). Inserting this expression in Eq. (15) we find that the only term of order $\mathcal{O}(1)$ comes from setting $i = 1, j = 2$ in the sum above. That is, at leading order,

$$\frac{d}{dt} D^2_{\text{Fish}}(\boldsymbol{p}, \boldsymbol{p} + \boldsymbol{d}) = -a_{1 \leftarrow 2} \frac{d_1^2}{p_1^2} + \mathcal{O}(\varepsilon). \quad (19)$$

Since $a_{1 \leftarrow 2} < 0$ we can always find a $\varepsilon$ small enough so that this quantity is strictly positive. Hence, for any non-Markovian dynamics there always exists $\boldsymbol{p}$ and $\boldsymbol{d}$ such that $D^2_{\text{Fish}}(\boldsymbol{p}, \boldsymbol{p} + \boldsymbol{d})$ locally increases, proving the Theorem. $\square$

Theorem 1 can be seen as a completion of Chentsov's Theorem, as it implies that not only the Fisher information decreases under Markovian evolutions, but also that an evolution contracting the Fisher distance between any two points has to be Markovian. The proof generalizes to the case of quantum dynamics in the canonical CP-divisibility framework [3], by considering a copy of the system on which the dynamics acts trivially, i.e., the evolution is given by $\mathcal{T}^{(t,0)} \otimes \mathbb{1}_N$ and the set of states is considered on the global bipartition (cf. Appendix D.1).

Interestingly, a similar theorem cannot hold for the trace-distance as one can explicitly construct non-Markovian evolutions that monotonically contract $D_{\text{Tr}}(\boldsymbol{p}, \boldsymbol{q})$ for any two points

(see Appendix A). This corroborates the interpretation of the Fisher metric as the canonical distance whose contractivity identifies stochastic-divisible maps. Still, the argument that lead to Thm. 1 has a shortcoming: even if non-Markovianity implies the dilation of the Fisher information, this is not sufficient to produce an operational witness. In fact, assume that at time $t$ there is a local dilation for two points close to $\tilde{\boldsymbol{p}}$. In order to observe it, one would need to initialise the system in the state $\boldsymbol{p}(0) = (T^{(t,0)})^{-1}\tilde{\boldsymbol{p}}$. If $\tilde{\boldsymbol{p}}$ is outside the image of $T^{(t,0)}$, this cannot be achieved physically. That is, the drawback of this approach is that the Fisher metric is point-dependent, and the witnessing point might be excluded by the dynamics $T^{(t,0)}$. On the other hand, the trace-distance is translational invariant, i.e., $D_{\mathrm{Tr}}(\boldsymbol{p}, \boldsymbol{q}) = |\boldsymbol{d}|$ where $\boldsymbol{d} = \boldsymbol{p} - \boldsymbol{q}$. Then, as soon as any two points $\boldsymbol{p}$ and $\boldsymbol{q}$ are increasing their trace-distance, in order to present an operational witness it is sufficient to consider a point $\boldsymbol{r}$ in the interior of the image of $T^{(t,0)}$ and $\varepsilon$ small enough for $\boldsymbol{r}_\varepsilon = \boldsymbol{r} + \varepsilon(\boldsymbol{p} - \boldsymbol{q})$ to be in the image as well. Then, $D_{\mathrm{Tr}}(\boldsymbol{r}, \boldsymbol{r}_\varepsilon) = \varepsilon D_{\mathrm{Tr}}(\boldsymbol{p}, \boldsymbol{q})$ and this increases by assumption. Moreover, if one adds a finite number of ancillary degrees of freedom on which the dynamics acts trivially, one can always find such two points for any non-Markovian evolution (see [5] and App A). A similar property does not hold for the Fisher distance, as it lacks translation invariance. More specifically,

**Theorem 2.** *No finite number $n$ of copies of the channel $T^{(t,0)}$ nor ancillary degrees of freedom of any dimension $M$ is enough to witness all non-Markovian evolutions via revivals of the Fisher distance between two initially prepared states.*

Specifically, given $n$ copies of the system, and an ancilla with arbitrary dimension $M$, the state space will be $\mathcal{S}(\mathbb{R}^{N^{\otimes n}} \otimes \mathbb{R}^M)$ and the dynamics acting on it $\bar{T}^{(t,0)} = T^{(t,0)\otimes n} \otimes \mathbb{1}_M$.

*Proof.* We constructively provide, for any $n$ and $M$, a counterexample in which all the states in the image of $\bar{T}^{(t,0)}$ continue decreasing their Fisher distance between time $t$ and $t + \delta t$, even if $T^{(t+\delta t,t)} \simeq \mathbb{1} + \delta t R^{(t)}$ is non-stochastic. Here we provide the proof for the single copy case, deferring the multiple cases one to Appendix D.2. The map we consider is then given by $\bar{T}^{(t,0)} = T^{(t,0)} \otimes \mathbb{1}_M$ and the rate matrix by $\bar{R}^{(t)} = \frac{\mathrm{d}}{\mathrm{d}t}\bar{T}^{(t,0)} = R^{(t)} \otimes \mathbb{1}_M$. Suppose now that there is a unique negative rate $a_{\tilde{i} \leftarrow \tilde{j}}$ and that the image of $T^{(t,0)}$ is contained to a small ball around an appropriate vector $\pi$ (e.g., by a map of the form $T^{(t,0)}[\boldsymbol{p}] = \pi(1-\varepsilon) + \varepsilon \boldsymbol{p}$).[4] Attach at time 0 an arbitrary ancilla, so that the initial state is given by $\boldsymbol{p}(0) \in \mathcal{S}(\mathbb{R}^N \otimes \mathbb{R}^M)$, and define $\boldsymbol{w}$ to be its reduced marginal on $\mathbb{R}^M$, whereas the dynamics is given by $\bar{T}^{(t,0)} = T^{(t,0)} \otimes \mathbb{1}_M$. Then, the state at time $t$ will be $\varepsilon$-close to

$$\boldsymbol{p}(t) \sim \pi \otimes \boldsymbol{w} + \mathcal{O}(\varepsilon). \tag{20}$$

Notice also that the rate matrix $\bar{R}^{(t)} = \frac{\mathrm{d}}{\mathrm{d}t}\bar{T}^{(t,0)} = R^{(t)} \otimes \mathbb{1}_M$ has the following coordinate expression

$$[\bar{R}]_{ij,\alpha\beta} = R_{ij}\delta_{\alpha\beta}, \quad i,j \in \{1,\ldots,N\}, \quad \alpha,\beta \in \{1,\ldots,M\}, \tag{21}$$

so that the rates are simply given by $a_{i\alpha \leftarrow j\beta} = a_{i \leftarrow j}\delta_{\alpha\beta}$. In this scenario, the evolution of the Fisher distance (as expressed in Eq. (15)) becomes

$$-\sum_{i \neq j, \alpha} a_{i \leftarrow j}\left(\frac{d_{i\alpha}}{p_{i\alpha}} - \frac{d_{j\alpha}}{p_{j\alpha}}\right)^2 p_{j\alpha} + \mathcal{O}(\varepsilon), \quad \text{with } p_{j\alpha} = \pi_j w_\alpha. \tag{22}$$

Again, consider the case in which at time $t$ a single rate becomes negative, for definiteness say $a_{1 \leftarrow 2} < 0$. Then, it is sufficient that $a_{2 \leftarrow 1}\pi_1 > |a_{1 \leftarrow 2}|\pi_2$ to see that the sum in Eq. (22) is strictly negative in the limit $\varepsilon \to 0$. Hence, even if the dynamics is non-Markovian, there is no increase in Fisher distance on the image of $T^{(t,0)}$, proving Theorem 2 for $n = 1$. $\qquad\square$

---

[4]This choice of counterexample allows us again to use the local expression of the Fisher metric (2) to prove global properties of the same distance.

In the same way, even using multiple copies of the channel does not help finding a witness. The proof easily generalises to $n \geq 2$, as presented in App. D.2. It should be noticed that the condition of finite copies in Thm. 2 cannot be dropped: in fact, in the limit of infinite copies, one can perform full tomography of the evolution, allowing to reconstruct its action also on points outside of the image of $T^{(t,0)}$.

Despite the above "no-go" Theorem, we can introduce an operational non-Markovianity witness which does not require additional copies of the channel, but only some specific post-processing of the states. More specifically, the following technical theorem holds

**Theorem 3.** *For any state $p$ and perturbation $d$ on $\mathcal{S}(\mathbb{R}^N \otimes \mathbb{R}^M)$ (where we admit an M-dimensional ancilla), it is possible to implement a class of transformations $F_d$ depending on $d$ and on $T^{(t,0)}$ that witness non-Markovianity at time t. That is, if $T^{(t+\delta t,t)}$ is stochastic, for any choice of $d$, one has that*

$$D_{\text{Fish}}\left(F_d \circ T^{(t+\delta t,0)}[p], F_d \circ T^{(t+\delta t,0)}[p+d]\right) \leq D_{\text{Fish}}\left(F_d \circ T^{(t,0)}[p], F_d \circ T^{(t,0)}[p+d]\right), \quad (23)$$

*whereas in the presence of non-Markovianity (i.e., for $T^{(t+\delta t,t)}$ not stochastic) there exists at least one $d$ for which the inequality is reversed (i.e., the Fisher distance of the post-processed states increases). Moreover, for classical systems $M = 2$ is enough to witness in this way all non-Markovian evolutions.*

The specific proof is given in Appendix D.3. Theorem 3 ensures that any break of stochastic-divisibility in the interval $[t, t + \delta t]$ can be operationally witnessed via backflow of Fisher information between states that undergo the transformation $F_d$ before being measured. The specific construction that we used here requires the knowledge of the previous dynamics $T^{(t,0)}$, which makes such witnessing unpractical. Still, this protocol should be considered as a proof of principle of the possibility of witnessing non-Markovianity through post-processing.

**A case study.** Theorem 1 tells us that any non-Markovian evolution induces the dilation of the Fisher distance somewhere in the space of states. On the other hand, as it was discussed above, this is not the case for another important information quantifier, namely the trace distance (1). In order to exemplify this behaviour, we present here a family of non-Markovian evolutions that cannot be witnessed on the space of states by the trace distance. In particular, consider the dynamics represented as

$$p(t) = (1 - s(t))p(0) + s(t)m(t), \quad (24)$$

where $0 \leq s(t) \leq 1$ is a smooth mixing parameter satisfying $s(0) = 0$, and $\dot{s}(t) \geq 0$, while $m(t)$ is a state that can vary in time. The master equation resulting from (24) is (with time-dependence implicit in the notation)

$$\dot{p} = -\frac{\dot{s}}{1-s}(p - m) + s\dot{m}. \quad (25)$$

This kind of dynamics is explicitly Markovian whenever the state $m$ is fixed. However if $\dot{m} \neq 0$ the evolution is not stochastic-divisible (i.e. non-Markovian) in general. In fact, one can express the rates $a_{i \leftarrow j}^{(t)}$ defined in (12) by simple algebraic manipulations as $a_{i \leftarrow j} = \frac{\dot{s}}{1-s}m_i + s\dot{m}_i$, which can be negative due to the second term.

If we now look at the trace distance, though, this is always decreasing, despite the non-Markovianity. In fact, it is clear from (24) that

$$|p(t) - q(t)| = (1 - s(t))|p(0) - q(0)|, \quad (26)$$

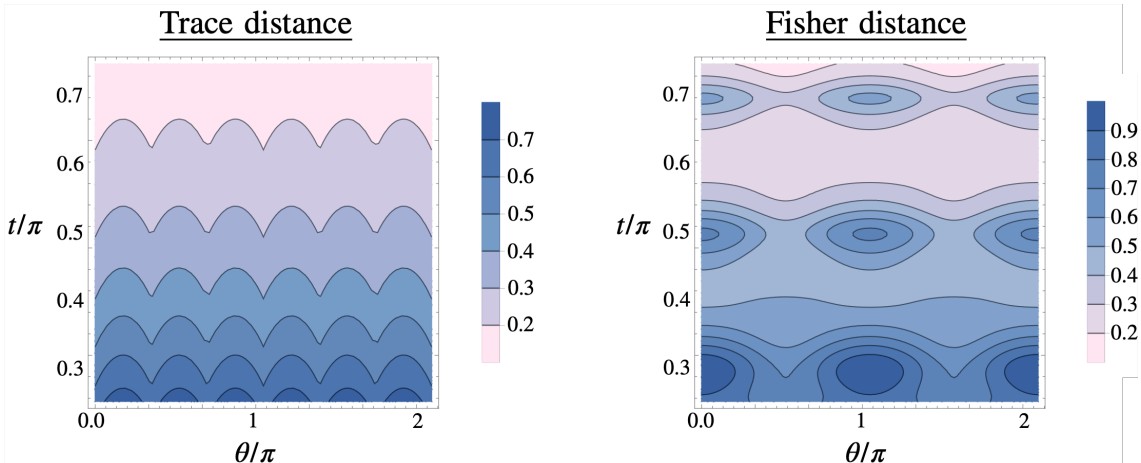

Figure 1: Evolution of the trace distance and of the Fisher distance between the points $\boldsymbol{p}(t)$ and $\boldsymbol{p}(t) + \varepsilon \boldsymbol{d}_\theta(t)$, specified by the dynamics (27), $\boldsymbol{p}(0) = \{\frac{1}{5}, \frac{2}{5}, \frac{2}{5}\}$ and $\boldsymbol{d}_\theta(0) := \frac{\cos\theta}{\sqrt{2}}\{0, 1, -1\} + \frac{\sin\theta}{\sqrt{6}}\{2, -1, -1\}$, which covers all possible directions in the tangent space. The time and the parameter $\theta$ are given in units of $\pi$, whereas the scale for the contour plot is given in units of $\varepsilon$. As it can be noticed, the trace distance does not witness the backflow (notice that since it is translational invariant, this continues to hold on the whole phase space). The Fisher distance, on the other hand, shows a non-monotonic behaviour in time.

which monotonically decreases, as we assumed that $\dot{s} \geq 0$. On the other hand, the Fisher distance can indeed detect non-Markovianity, as follows from Theorem 1. This difference in behaviour is presented in Fig. 1, where we plot the evolution of both trace distance and Fisher distance between two infinitesimally-close states of a three level system, whose dynamics is a specific example of (24):

$$\boldsymbol{p}(t) = e^{-t}\boldsymbol{p}(0) + \left(\frac{1 - e^{-t}}{2}\right)\big((1 + \cos(10t))\boldsymbol{v}_1 + (1 - \cos(10t))\boldsymbol{v}_2\big), \qquad (27)$$

where $\boldsymbol{v}_1 := \{\frac{1}{3}, \frac{1}{3}, \frac{1}{3}\}$ and $\boldsymbol{v}_2 = \{1, 0, 0\}$.[5] Thanks to the fact that the tangent space of $\mathcal{S}(\mathbb{R}^3)$ is two-dimensional, we can plot all of it (up to normalization), showing how there is no direction for which the trace distance is increasing. The Fisher distance, instead, which is plotted on the right side of the figure, shows a sequence of periodic backflows in time, as one could expect from the equation of motion.

## 4 Backflows of information from Bayesian retrodiction

In most of the literature about non-Markovianity backflows of information are considered by studying states $\boldsymbol{p}(t)$ at time $t$ [3,4], while the question about backflows of information *about the initial state* $\boldsymbol{p}(0)$ remains instead largely unexplored. Even if one can argue that the invertibility of the dynamics preserves the information about the initial conditions (as these can be recovered using $\boldsymbol{p}(0) = T^{(t,0)^{-1}}[\boldsymbol{p}(t)]$), actually retrieving the initial state from $\boldsymbol{p}(t)$ requires full tomography both of the state and of the channel, and a post-processing of such data which cannot be performed physically in a single-shot scenario.

---

[5]One can interpret the dynamics (27) as a time dependent multi-level amplitude damping channel [25, 26], with a thermal fixed point oscillating between high temperature ($\boldsymbol{v}_1$) and low temperature ($\boldsymbol{v}_2$).

Conversely, we consider here a Bayesian inversion of $p(t)$ that allows us to compare it with the initial state through a physically implementable transformation. In particular, define the *prior* as a vector $\pi$ representing our knowledge about the system at time 0. Given an evolution $T^{(t,0)}$, the Bayes-recovery map is defined as

$$\hat{T}_t = J_\pi \circ T_t^\mathsf{T} \circ J_{T[\pi]}^{-1}, \tag{28}$$

where we use the shorthand notation $T_t := T^{(t,0)}$, and we introduced the map $J_p$, corresponding to the diagonal operator that multiplies each component of a vector by the corresponding component of $p$, i.e. $[J_p]_{ij} = \delta_{ij} p_j$.

The map $\hat{T}_t$ is stochastic (i.e., physically implementable) and represents a recovery of the state via statistical retrodiction [27, 28]. Indeed if one identifies the coordinates of the prior with the corresponding probability, i.e., $\pi_i \equiv P(i, 0)$, and the components of the maps with the corresponding transitions, i.e., $(T_t)_{ij} \equiv P(i, t|j, 0)$, it is easy to verify that $(\hat{T}_t)_{ij} \equiv P(i, 0|j, t)$ satisfies the Bayes rule. It should also be noticed that $\hat{T}_t$ perfectly recovers the prior at all times ($\pi = \hat{T}_t \circ T_t[\pi]$).

This channel allows us to study how much information is stored in the evolved state $p(t)$ about its initial conditions. In particular, we can compare the distance between $p(0)$ and the retrodicted state $\hat{p}(t) := \hat{T}_t[p(t)] = \hat{T}_t \circ T_t[p(0)]$. We also assume that the prior contains some knowledge on the initial conditions, so that we can write $p(0) = \pi + d$, for some small $|d| \ll 1$. Moreover, since $\pi$ is perfectly recovered, we also have that $\hat{p}(t) = \pi + \hat{d}(t)$, and $p(0) - \hat{p}(t) = d - \hat{d}(t)$ is also infinitesimal. Then, the Fisher information (2) between $p(0)$ and $\hat{p}(t)$ reads, up to order $\mathcal{O}(|d|^2)$,

$$D_{\text{Fish}}^2(p(0), \hat{p}(t)) \simeq \left\langle d - \hat{d}(t), d - \hat{d}(t) \right\rangle_{p(0)} \simeq \left\langle d - \hat{d}(t), d - \hat{d}(t) \right\rangle_\pi. \tag{29}$$

Interestingly, this object is directly connected to the Fisher information at time $t$:

**Theorem 4.** *The contractivity of the Fisher information at time $t$ is in one-to-one correspondence with the expansivity of Eq.* (29). *That is,*

$$\frac{\mathrm{d}}{\mathrm{d}t} D_{\text{Fish}}^2(p(0), \hat{p}(t)) = \frac{\mathrm{d}}{\mathrm{d}t} \left\langle d - \hat{d}(t), d - \hat{d}(t) \right\rangle_\pi \geq 0, \tag{30}$$

*if and only if the Fisher information contracts for any two points in the vicinity of $T_t[\pi]$.*

*Proof.* The main ingredient in the proof of this theorem is given by the following identities

$$\langle d, \hat{T}_t T_t[d] \rangle_\pi = \langle T_t[d], T_t[d] \rangle_{T_t[\pi]} = \langle \hat{T}_t T_t[d], d \rangle_\pi, \tag{31}$$

which can verified by directly substituting $\hat{T}_t$ with its definition in Eq. (28). One can read from these equalities the following two facts: first, $\hat{T}_t$ can be used to put in relation the Fisher information at time 0 and at time $t$; secondly, $\hat{T}_t T_t$ is self-adjoint with respect to $\langle \bullet, \bullet \rangle_\pi$. This allows to rewrite Eq. (30) as

$$\frac{\mathrm{d}}{\mathrm{d}t} \left\langle d, (\mathbb{1} - \hat{T}_t T_t)^2[d] \right\rangle_\pi = -2 \left\langle d, (\mathbb{1} - \hat{T}_t T_t) \frac{\mathrm{d}}{\mathrm{d}t} \hat{T}_t T_t[d] \right\rangle_\pi. \tag{32}$$

First notice that $(\mathbb{1} - \hat{T}_t T_t)$ is positive definite. In fact, this can be seen from

$$\langle d, \hat{T}_t T_t[d] \rangle_\pi = \langle T_t[d], T_t[d] \rangle_{T_t[\pi]} \leq \langle d, d \rangle_\pi, \tag{33}$$

where the last inequality follows from the contractivity of the Fisher information. Moreover, we also have that $\langle d, \frac{\mathrm{d}}{\mathrm{d}t} \hat{T}_t T_t[d] \rangle_\pi = \frac{\mathrm{d}}{\mathrm{d}t} \langle T_t[d], T_t[d] \rangle_{T_t[\pi]}$, so that $-\frac{\mathrm{d}}{\mathrm{d}t} \hat{T}_t T_t$ is positive if and only

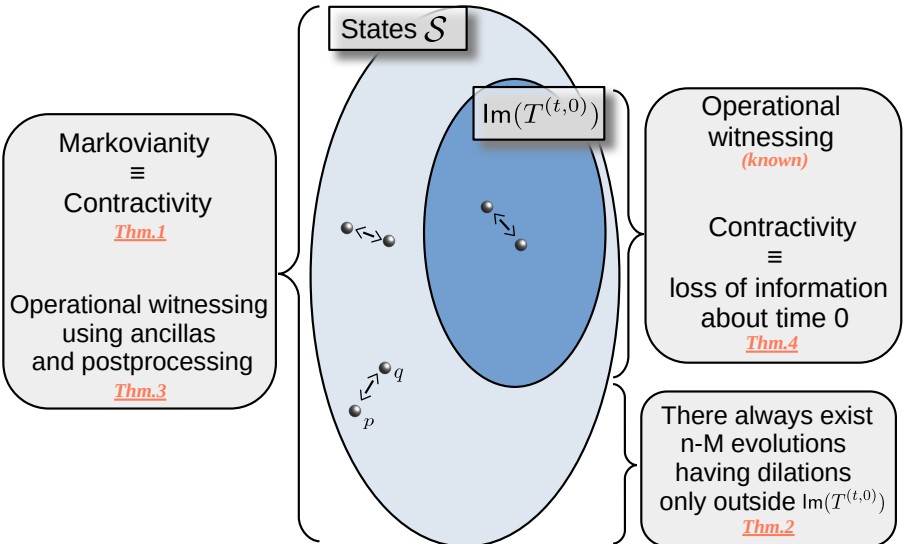

Figure 2: The main quantity analyzed in this work is the rate of contraction/dilation of the Fisher metric, i.e., $\frac{d}{dt}D_{\text{Fish}}(\boldsymbol{p},\boldsymbol{q})$, with $|\boldsymbol{p}-\boldsymbol{q}| \ll 1$, when both $\boldsymbol{p}$ and $\boldsymbol{q}$ evolve according to the local intermediate map $T^{(t+\delta t,t)}$. We characterize the mathematical and operational meaning of the negativity/positivity of such rate, both inside and outside the image of the evolution $T^{(t,0)}$.

if the Fisher contracts monotonically. If this is the case, we can use the fact that the product of two positive operators has positive spectrum. On the other hand, if the Fisher metric is expanding at time $t$, there exists an eigenvector $\tilde{\boldsymbol{d}}$ of $-\frac{d}{dt}\hat{T}_t T_t$ with negative eigenvalue $\lambda < 0$, so that

$$-2\left\langle \tilde{\boldsymbol{d}}, (\mathbb{1}-\hat{T}T)\frac{d}{dt}\hat{T}T[\tilde{\boldsymbol{d}}]\right\rangle_\pi = 2\lambda\left\langle \tilde{\boldsymbol{d}}, (\mathbb{1}-\hat{T}T)[\tilde{\boldsymbol{d}}]\right\rangle_\pi < 0, \tag{34}$$

proving the claim. □

This theorem tells us that the ability of an agent of retrieving the initial state of the dynamics decreases under Markovian evolution, as it corresponds to the monotonous contraction of the Fisher information. Viceversa, in the case in which there are backflows in the Fisher information, non-Markovianity helps obtaining more information about the initial state.

We notice that, contrary to the Theorems presented in Sec. 3, Theorem 4 holds only locally, i.e., assuming that the difference $\boldsymbol{d}$ between prior and the state at time $t = 0$ is small. Moreover, the generalization of Theorem 4 to the quantum scenario requires a technical analysis of quantum Bayes retrodiction [28], and will be presented in a forthcoming work [29].

## 5 Conclusions

In this work we characterized the relation between Markovianity, Fisher metric contractivity and information flow, both from the mathematical and operational point of view. The resulting picture can be seen in Fig. 2: we showed that monotonous contractivity of the Fisher metric on the whole set of states and at all times, is mathematically equivalent to Markovianity (Thm. 1). As known, when the metric dilates locally inside the image of the evolution $\text{Im}(T^{(t,0)})$, a backflow of Fisher information can be operationally witnessed. At the same time, non-Markovian

evolutions might in general show Fisher metric dilations only outside the image of the evolution itself, regardless of the number of copies of the channel and ancillary degrees of freedom available (Thm. 2). To witness operationally non-Markovianity in such cases, one needs post-processing to be appended to the dynamics (Thm. 3). Finally, we showed that dilations of the Fisher metric between evolving states can be mapped to a backflow of information about the initial states by applying Bayesian retrodiction (Thm. 4). Our work therefore corroborates the idea that the Fisher Information defines a natural metric on the space of states, as its contractivity properties characterize memory effects in open system dynamics, both from the mathematical and operational point of view.

## 5.1 The case of quantum dynamics

The results were presented for classical dynamics in order to keep the exposition clean. In fact, whereas the Fisher information distance is uniquely defined for classical systems, the presence of non commutative observables allows only for the definition of a family of Fisher information in the quantum case (see Appendix C). Despite this technical difficulty, we also show in Appendix C that in order to study the interplay between Markovianity and Fisher information we can limit ourselves to the diagonal subspaces, practically going back to the classical constructions (details for each case are given in Appendix D). In this way, Theorems 1, 2 and 3 can be seen to hold quantum scenario. The only case in which we cannot ignore the non-uniqueness of the quantum Fisher information are the expression of the Fisher information flows Eq. (16), and Thm. 4. Both of these results will be generalised to the quantum regime in the forthcoming work [29].

Moreover, it should be noticed that for quantum dynamics we identify Markovianity with CP-divisibility [3]. This enforces an important difference with the classical case: whereas for the classical dynamics the minimal space that can be used to probe the Markovianity of an $N$-dimensional system is given by $\mathcal{S}(\mathbb{R}^N)$, the corresponding for quantum systems is given by two copies of the same Hilbert space on which the dynamics acts as $\mathcal{T}^{(t,0)} \otimes \mathbb{1}_N$. In fact, it is not possible to distinguish P-divisibility from CP-divisibility by restricting to the system space alone without the use of any ancillas.

# Acknowledgements

**Funding information** The authors acknowledge support by the Government of Spain (FIS2020-TRANQI and Severo Ochoa CEX2019-000910-S), Fundacio Cellex, Fundacio Mir-Puig, Generalitat de Catalunya (CERCA, AGAUR SGR 1381). PA is supported by "la Caixa" Foundation (ID 100010434, Grant No. LCF/BQ/DI19/11730023). MS is supported by European Union's Horizon 2020 research and innovation programme under the Marie Skłodowska-Curie Grant No. 713729. DDS is supported by ERC AdG CERQUTE.

# A The trace distance: contractivity and witnesses

The trace distance between two classical states is given by $|\boldsymbol{p} - \boldsymbol{q}|_{\mathrm{Tr}} \equiv \sum |p_i - q_i|$. Interestingly, it only depends on the vector $\boldsymbol{d} := \boldsymbol{q} - \boldsymbol{p}$ and not on the base-point $\boldsymbol{p}$. Distances satisfying this condition are called *translational invariant*. Then, we can write its evolution as:

$$\frac{\mathrm{d}}{\mathrm{d}t} D_{\mathrm{Tr}}(\boldsymbol{p},\boldsymbol{q}) = \frac{\mathrm{d}}{\mathrm{d}t} |\boldsymbol{d}|_{\mathrm{Tr}} = \sum_i \frac{\mathrm{d}}{\mathrm{d}t} |d_i| = \sum_i \mathrm{sign}(d_i) \dot{d}_i . \tag{A.1}$$

For classical systems, given a rate matrix decomposed as in Eq. (12), one has that

$$\dot{d}_i = \sum_j R_{i,j} d_j = \sum_{\substack{j \\ j \neq i}} \left( a_{i \leftarrow j} d_j - a_{j \leftarrow i} d_i \right), \tag{A.2}$$

so that we can rewrite Eq. (A.1) as

$$\sum_{i,j,i\neq j} \text{sign}(d_i) \left( a_{i\leftarrow j} d_j - a_{j\leftarrow i} d_i \right) = \sum_{i,j,i\neq j} \left( \text{sign}(d_j) a_{j\leftarrow i} d_i - \text{sign}(d_i) a_{j\leftarrow i} d_i \right) \tag{A.3}$$

$$= \sum_{i,j,i\neq j} \left( \text{sign}(d_j) - \text{sign}(d_i) \right) a_{j\leftarrow i} d_i \leq 0, \tag{A.4}$$

where the last expression is clearly negative whenever $a_{j \leftarrow i}$ are positive: in fact, either $d_j$ and $d_i$ have equal sign, in which case the term is null, or $\text{sign}(d_j) = -\text{sign}(d_i)$, so that the factor $(\text{sign}(d_j) - \text{sign}(d_i)) = -2\,\text{sign}(d_i)$, and we can rewrite the terms in the sum as $-2\,\text{sign}(d_i)d_i = -2|d_i|$.

This shows explicitly how the trace distance decreases under Markovian maps. Is the reverse true? That is, if the evolution stops being stochastic-divisible, i.e., some $a_{\tilde{i}\leftarrow\tilde{j}}$ is negative, can one always find two points for which the trace distance between them increases? Differently from the Fisher distance, the answer to this question is negative: one easy counterexample can be given in dimension $N = 2$, with $a_{1\leftarrow 2} < 0$ and $a_{2\leftarrow 1} > 0$ with $|a_{2\leftarrow 1}| > |a_{1\leftarrow 2}|$. Plugging such choice in Eq. (A.4), and using the fact that in dimension 2, $\boldsymbol{d}$ satisfies $d_1 = -d_2$, it is easy to check that the derivative of $|\boldsymbol{d}|_{\text{Tr}}$ stays negative.

This fact could sound counter-intuitive: it is well-known that a map is stochastic *if and only if* each vector $\boldsymbol{v}$ decreases in trace-norm [3]. Yet, we have proven here that the trace distance cannot witness all non-Markovian evolutions, without resorting to ancillas. An easy resolution of this paradox is given by noticing that the test vectors used here are all in the form $\boldsymbol{d} = \boldsymbol{p} - \boldsymbol{q}$, constraining them to be of zero trace. This lowers the dimension of the tested vectors by one, resolving the contradiction. It is in fact straightforward to verify that simply by choosing $v_i = \delta_{i,2}$ in our example above one would be able to witness non-Markovianity (where $\delta_{i,j}$ is the Kronecker delta). Still, the trace condition prevents us from accessing this vector.

If one allows for the use of ancillas, though, the situation changes. In fact, consider an ancillary set $\mathcal{S}(\mathbb{R}^M)$ on which the dynamics acts trivially (i.e., states belong to $\mathcal{S}(\mathbb{R}^N \otimes \mathbb{R}^M)$ and the dynamics is of the form $\bar{T}^{(t,0)} = T^{(t,0)} \otimes \mathbb{1}_M$). It is now sufficient to consider the following vector on the extended space

$$\boldsymbol{d} = \boldsymbol{v} \otimes \boldsymbol{d}_{\text{anc}}, \quad \boldsymbol{v} \in \mathbb{R}^N, \quad v_i = \delta_{i,2}, \quad \boldsymbol{d}_{\text{anc}} \in \mathbb{R}^M, \quad \text{Tr}[\boldsymbol{d}_{\text{anc}}] = 0, \tag{A.5}$$

to witness the non-Markovianity in the example above. Notice that the trace condition on $\boldsymbol{d}_{\text{anc}}$ ensures that $\boldsymbol{d}$ is traceless, so this is a valid distance vector. Then, since its derivative takes the form $\dot{\boldsymbol{d}} = \dot{\boldsymbol{v}} \otimes \boldsymbol{d}_{\text{anc}}$, the trace distance increases in time

$$\frac{\mathrm{d}}{\mathrm{d}t}|\boldsymbol{d}| = |\dot{\boldsymbol{v}} \otimes \boldsymbol{d}_{\text{anc}}| = |\dot{\boldsymbol{v}}||\boldsymbol{d}_{\text{anc}}| > 0, \tag{A.6}$$

as $\boldsymbol{v}$ was chosen to have $|\dot{\boldsymbol{v}}| > 0$ when $a_{1\leftarrow 2} < 0$. Notice that the ancilla can have dimension as small as 2. The same kind of reasoning is used in Ref. [5] for the case of quantum dynamics. From these simple considerations and from reference [5] we have the following

**Lemma 1.** *Given a non-stochastic-divisible dynamics $T^{(t,0)}$, adding a finite ancilla and considering the dynamics $T^{(t,0)} \otimes \mathbb{1}_M$ allows for witnessing any such dynamics via revivals in trace distance between initially prepared states. A finite number $M$ of ancillary degrees of freedom is enough, and in the classical case $M = 2$.*

In the quantum case, $M = N + 1$ is needed, where $N$ is the dimension of the quantum state [5]. We close the section with a final remark. Notice that even *less* than a finite ancilla would be enough. By this we mean that it is enough to enlarge the state space with an additional microstate that does not interact with the others, i.e., considering $\mathcal{S}(\mathbb{R}^N) \to \mathcal{S}(\mathbb{R}^{N+1})$ and a dynamics of the form $T' = T \oplus 1$ (i.e., $T'_{ij} = T_{ij}$ for $i, j \in \{1, \dots, N\}$ and $T'_{ij} = \delta_{ij}$ if one between $i$ and $j$ is the $N+1$ index). Given $\boldsymbol{v} \in \mathbb{R}^N$ such that $|\dot{\boldsymbol{v}}| > 0$, one can consider $\boldsymbol{d} \in \mathbb{R}^{N+1}$ with

$$
\boldsymbol{d} = \begin{cases} d_i = v_i, & \text{if } i \in \{1, \dots, N\}, \\ d_i = -\sum_{i=1}^{N} v_i, & \text{if } i \equiv N+1. \end{cases} \tag{A.7}
$$

Since the map $T$ is trace-preserving at all times, such vector satisfies

$$
|\dot{\boldsymbol{d}}| = |\dot{\boldsymbol{v}}|, \tag{A.8}
$$

allowing for the witnessing of non-Markovianity. This result should be compared with the discussion above about how the trace condition lowers the accessible dimension by one.

## B   Rates of Fisher Information flow and contractivity

We present here a decomposition of the derivative of the Fisher information in a sum of independent flows. In particular, by explicitly using the expression in Eq. (12) for the dynamics we can compute

$$
\begin{aligned}
2 \frac{\mathrm{d}}{\mathrm{d}t} D_{\text{Fish}}^2(\boldsymbol{p}, \boldsymbol{p} + \boldsymbol{d}) &= \sum_i \left( \frac{2 d_i \dot{d}_i}{p_i} - \frac{d_i^2}{p_i^2} \dot{p}_i \right) \\
&= \sum_{i,j} \left( \frac{2 d_i (a_{i \leftarrow j} d_j - a_{j \leftarrow i} d_i)}{p_i} - \frac{d_i^2}{p_i^2} (a_{i \leftarrow j} p_j - a_{j \leftarrow i} p_i) \right) \\
&= \sum_{i,j} \left( \frac{2 d_i d_j a_{i \leftarrow j}}{p_i} - \frac{d_i^2}{p_i} a_{j \leftarrow i} - \frac{d_i^2}{p_i^2} p_j a_{i \leftarrow j} \right) \\
&= \sum_{i,j} a_{i \leftarrow j} \left( \frac{2 d_i d_j}{p_i} - \frac{d_j^2}{p_j} - \frac{d_i^2}{p_i^2} p_j \right) \\
&= -\sum_{i \neq j} a_{i \leftarrow j} \left( \frac{d_i}{p_i} - \frac{d_j}{p_j} \right)^2 p_j \le 0, \tag{B.1}
\end{aligned}
$$

where between the third and the fourth line we made the change of variable $i \to j$ in order to factor out $a_{i \leftarrow j}$. Whenever the evolution is Markovian (i.e., the rates $a_{i \leftarrow j}$ are all positive), the Fisher information is contracting, in analogy with what happened for the trace distance. On the other hand, if one compares the result in Eq. (A.4) with what we just obtained, there is an important difference between the two: whereas the evolution of the trace distance only depends on $\boldsymbol{d}$ (mirroring the translational invariance of this quantity), the Fisher information explicitly depends on the base-point. This difference is particularly important when considering the technical proofs of the results of this work.

## C  The quantum Fisher information metric and its relation to the trace distance

The extension of the Fisher information to quantum systems is done by generalising Chentsov's theorem to completely positive, trace preserving maps (CPTP). That is, a metric on quantum states is called monotone if it decreases under all CPTP maps. Then, it was shown by Petz in [23] that all such metrics are induced by scalar products of the form:

$$K_\rho^f(A,B) := \frac{1}{2}\mathrm{Tr}\left[A^\dagger \, \mathbb{J}_{f,\rho}^{-1}[B]\right], \tag{C.1}$$

where $\mathbb{J}_{f,\rho}$ is a self-adjoint superoperator given by:

$$\mathbb{J}_{f,\rho} := \mathbb{R}_\rho \, f\left(\mathbb{L}_\rho \mathbb{R}_\rho^{-1}\right), \tag{C.2}$$

and $\mathbb{L}_\rho/\mathbb{R}_\rho$ are the left/right multiplication operators acting as $\mathbb{L}_\rho \pi = \rho \pi$ (respectively $\mathbb{R}_\rho \pi = \pi \rho$) and $f : \mathbb{R}^+ \to \mathbb{R}^+$ is a standard operator monotone function. Despite its complicated form, the interpretation of $K_\rho^f(A,B)$ as the natural extension of the Fisher information to the quantum regime is corroborated by the result in [30], where it was shown that the same quantity emerges from the local expansion of the quantum generalisation of Csizár divergences. For this reason, we define the family of quantum Fisher distances as

$$D_{\mathrm{Fish},f}^2(\rho, \rho + \delta\rho) \simeq K_\rho^f(\delta\rho, \delta\rho) := \frac{1}{2}\mathrm{Tr}\left[\delta\rho^\dagger \, \mathbb{J}_{f,\rho}^{-1}[\delta\rho]\right]. \tag{C.3}$$

The uniqueness of the classical Fisher metric is hence substituted with a whole family of different monotone metrics. It is interesting to point out, though, that when $\mathbb{J}_{f,\rho}$ acts on diagonal states it behaves (independently of $f$) as the multiplication by $\rho$. That is, if $[A,\rho] = [B,\rho] = 0$, with a small abuse of notation we have

$$K_\rho^f(A,B) := \frac{1}{2}\mathrm{Tr}\left[\frac{A^\dagger B}{\rho}\right] = \langle A,B\rangle_\rho \,, \tag{C.4}$$

that is all the quantum Fisher metrics collapse to the classical one for diagonal states. This feature allows us to lift most of our classical constructions to the quantum scenario without further complications.

Specifically, the following Lemma will be particularly relevant:

**Lemma 2.** *Given a state $\rho = \sum_i \rho_i |i\rangle\langle i|$, any perturbation $\delta\rho$ can be decomposed in diagonal and coherent part as:*

$$\delta\rho = \delta\rho_\Delta + \delta\rho_C, \quad \text{with} \quad [\rho, \delta\rho_\Delta] = 0 \quad \text{and} \quad \langle i|\delta\rho_C|i\rangle = 0, \tag{C.5}$$

*that is, $\delta\rho_\Delta$ is the diagonal part of the matrix $\delta\rho$ and $\delta\rho_C$ the off-diagonal. Then for all $f$*

$$\mathrm{Tr}\left[\delta\rho \, \mathbb{J}_{f,\rho}^{-1}[\delta\rho]\right] = \mathrm{Tr}\left[\delta\rho_\Delta \, \mathbb{J}_{f,\rho}^{-1}[\delta\rho_\Delta]\right] + \mathrm{Tr}\left[\delta\rho_C \, \mathbb{J}_{f,\rho}^{-1}[\delta\rho_C]\right] \tag{C.6}$$

$$= 2\langle\boldsymbol{\delta},\boldsymbol{\delta}\rangle_{\boldsymbol{\rho}} + \mathrm{Tr}\left[\delta\rho_C \, \mathbb{J}_{f,\rho}^{-1}[\delta\rho_C]\right], \tag{C.7}$$

*where the components of the vectors $\boldsymbol{\delta}$, $\boldsymbol{\rho}$ are specified as*

$$\boldsymbol{\rho}_i = \langle i|\rho|i\rangle, \quad \boldsymbol{\delta}_i = \langle i|\delta\rho|i\rangle = \langle i|\delta\rho_\Delta|i\rangle. \tag{C.8}$$

This result directly follows from the fact that

$$\mathrm{Tr}\left[\delta\rho_\Delta\, \mathbb{J}_{f,\rho}^{-1}[\delta\rho_C]\right] = \mathrm{Tr}\left[\delta\rho_C\, \mathbb{J}_{f,\rho}^{-1}[\delta\rho_\Delta]\right] = 0, \tag{C.9}$$

since $\mathbb{J}_{f,\rho}^{-1}[\delta\rho_\Delta]$ is itself diagonal in the basis $|i\rangle$ of eigenvectors of $\rho$. Then, we can use the Lemma above to prove the following corollary:

**Corollary 1.** *Consider a perturbation of the form $\delta\rho = \delta\rho_\Delta + \mathrm{d}t\,\delta\rho_C$, where $\mathrm{d}t$ is an infinitesimal quantity. Then, from Lemma 2 it follows that*

$$\mathrm{Tr}\left[\delta\rho\, \mathbb{J}_{f,\rho}^{-1}[\delta\rho]\right] = \mathrm{Tr}\left[\delta\rho_\Delta\, \mathbb{J}_{f,\rho}^{-1}[\delta\rho_\Delta]\right] + \mathcal{O}\left(\mathrm{d}t^2\right) = 2\,\langle\boldsymbol{\delta},\boldsymbol{\delta}\rangle_\rho + \mathcal{O}\left(\mathrm{d}t^2\right). \tag{C.10}$$

*In particular, the time derivative of the Fisher Information between $\rho$ and $\rho+\delta\rho$ for $[\rho,\delta\rho]=0$ coincides with the derivative of the classical Fisher Information. That is:*

$$\frac{1}{2}\mathrm{Tr}\left[\delta\rho\, \mathbb{J}_{f,\rho}^{-1}[\delta\rho]\right] = \langle\boldsymbol{\delta},\boldsymbol{\delta}\rangle_\rho \quad and \quad \frac{1}{2}\frac{\mathrm{d}}{\mathrm{d}t}\mathrm{Tr}\left[\delta\rho\, \mathbb{J}_{f,\rho}^{-1}[\delta\rho]\right] = \frac{\mathrm{d}}{\mathrm{d}t}\langle\boldsymbol{\delta},\boldsymbol{\delta}\rangle_\rho. \tag{C.11}$$

In fact, consider the scenario in which initially the perturbation is of the form $\delta\rho \equiv \delta\rho_\Delta$. In order to compute the derivative, one considers the evolution of the state $\rho+\delta\rho$ for a time $\mathrm{d}t$, which we denote by $\tilde{\rho}+\delta\tilde{\rho}$. Then, the perturbation has the form $\delta\tilde{\rho} = \delta\tilde{\rho}_\Delta + \mathrm{d}t\,\delta\tilde{\rho}_C$, so that we are in the situation of Eq. (C.10) (notice that it doesn't matter whether we take $\delta\tilde{\rho}_\Delta$ to be diagonal with respect to $\rho$ or to $\tilde{\rho}$, as this difference only contributes to order $\mathcal{O}(\mathrm{d}t)$). Since quadratic terms in $\mathrm{d}t$ do not contribute to the derivative, these considerations prove Eq. (C.11).

Finally, in the next Lemma we show that there are special points for which the trace distance and the Fisher information locally coincide. This result allows us to lift the many constructions present in the literature for the trace distance to the study of the Fisher information metric.

**Lemma 3.** *Choose an arbitrary perturbation $\delta\rho$. Then, consider the state $\rho_{\delta\rho} = \frac{|\delta\rho|}{\mathrm{Tr}[|\delta\rho|]}$. It holds that*

$$D_{\mathrm{Fish}}^2(\rho_{\delta\rho}, \rho_{\delta\rho}+\delta\rho) = \frac{1}{2}D_{\mathrm{Tr}}^2(\rho_{\delta\rho}, \rho_{\delta\rho}+\delta\rho) = \frac{1}{2}\mathrm{Tr}\left[|\delta\rho|\right]^2. \tag{C.12}$$

*Moreover, since $[\rho_{\delta\rho},\delta\rho]=0$, one can use Corollary 1 to show that:*

$$\frac{\mathrm{d}}{\mathrm{d}t}D_{\mathrm{Fish}}^2(\rho_{\delta\rho}, \rho_{\delta\rho}+\delta\rho) = \frac{1}{2}\frac{\mathrm{d}}{\mathrm{d}t}D_{\mathrm{Tr}}^2(\rho_{\delta\rho}, \rho_{\delta\rho}+\delta\rho) = \frac{1}{2}\frac{\mathrm{d}}{\mathrm{d}t}\mathrm{Tr}\left[|\delta\rho|\right]^2. \tag{C.13}$$

In fact, since $[\rho_{\delta\rho},\delta\rho] = 0$, the quantum Fisher information and its derivative can be studied just by looking at the quantity $\langle\boldsymbol{\delta},\boldsymbol{\delta}\rangle_{\rho_{\delta\rho}}$. But this is given by:

$$2\,\langle\boldsymbol{\delta},\boldsymbol{\delta}\rangle_{\rho_{\delta\rho}} = \sum_i \frac{\delta_i^2}{\left(\frac{|\delta_i|}{\mathrm{Tr}[|\delta\rho|]}\right)} = \sum_i \frac{\delta_i^2}{|\delta_i|}\sum_j |\delta_j| = \left(\sum_i |\delta_i|\right)^2 = D_{\mathrm{Tr}}^2\left(\rho_{\delta\rho}, \rho_{\delta\rho}+\delta\rho\right). \tag{C.14}$$

The fact that not only one can locally identify the Fisher distance and the trace distance, but also their first derivatives will be of key importance in many of our derivations.

# D  Additional proofs

## D.1  Theorem 1: quantum case

In the case of a quantum map $\mathcal{T}^{(t,0)}$ non-Markovianity means that $\mathcal{T}^{(t,0)}$ is not CP-divisible. This implies that there exists a time $t$ for which the Choi state $\mathcal{T}^{(t+dt,t)} \otimes \mathbb{1}_N[\,|\psi^+\rangle\langle\psi^+|\,]$ develops some negative eigenvalue (where $|\psi^+\rangle$ is the maximally entangled state). Call $|v\rangle$ the corresponding eigenvector. Since $\mathcal{T}^{(t+dt,t)}$ is infinitesimal, the Choi state must be close to $|\psi^+\rangle\langle\psi^+|$. But then, in order for $|v\rangle$ to correspond to a negative eigenvalue it must contain a non-zero component $|v_\perp\rangle$ orthogonal to $|\psi^+\rangle$. To see this, assume the opposite, i.e., $|v\rangle \equiv |\psi^+\rangle$ (as it is parallel to $|\psi^+\rangle$ and normalised); then, from perturbation theory we know that the corresponding eigenvalue must be $1 + \mathcal{O}(dt) > 0$. This contradicts the assumption that $|v\rangle$ is associated to a negative eigenvalue.

Consider now the state $\rho = |\psi^+\rangle\langle\psi^+|$ and the perturbation $\delta\rho = \varepsilon(|v_\perp\rangle\langle v_\perp| - |\psi^+\rangle\langle\psi^+|)$. With this choice we have that $[\rho, \delta\rho] = 0$. Moreover, the evolution of $\delta\rho$ through $\mathcal{T}^{(t+dt)}$ can only generate an off-diagonal component of order $\mathcal{O}(dt)$. Hence, we are in the situation of Corollary 2, that is we can neglect the off-diagonal contributions completely. In this way, we can simulate the process with a classical dynamics by only considering transitions from the diagonal into itself. Then, it is enough to notice that the classical rate $a_{v_\perp \leftarrow \psi^+}$ is negative by assumption. This concludes the proof.

## D.2  Theorem 2: many copy case

Consider the dynamics $\bar{T}^{(t,0)} = T^{(t,0)\otimes n} \otimes \mathbb{1}_M$ acting on the state space $\mathcal{S}(\mathbb{R}^{N\otimes n} \otimes \mathbb{R}^M)$. The rate matrix in this case is given by

$$\bar{R}^{(t)} = \frac{\mathrm{d}}{\mathrm{d}t}\,\bar{T}^{(t,0)} = \sum_{l=1}^{n} \mathbb{1}_N^{\otimes(l-1)} \otimes R^{(t)} \otimes \mathbb{1}_N^{\otimes(n-l)} \otimes \mathbb{1}_M\,. \tag{D.1}$$

Denoting by latin letters the indexes on the copies of the system space, and by greek letters the indexes of the ancillary space, we can express the rates as

$$\bar{a}^{(t)}_{(i_1 i_2 \dots i_n, \alpha) \leftarrow (j_1 j_2 \dots j_n, \beta)} = \sum_{l=1}^{n} a^{(t)}_{i_l \leftarrow j_l} \left( \delta_{i_1 j_1} \dots \delta_{i_{l-1} j_{l-1}} \delta_{i_{l+1} j_{l+1}} \dots \delta_{i_n j_n} \delta_{\alpha\beta} \right)\,. \tag{D.2}$$

In analogy with the construction for a single copy, it is now sufficient to consider a dynamics $T^{(t,0)}$ such that the image of $T^{(t,0)}$ is contained in a small ball around an appropriate $\pi$, i.e.,

$$\boldsymbol{p}(t) \sim \boldsymbol{\pi}^{\otimes n} \otimes \boldsymbol{q} + \mathcal{O}(\varepsilon)\,, \quad \boldsymbol{\pi} \in \mathcal{S}\left(\mathbb{R}^N\right)\,, \quad \boldsymbol{q} \in \mathcal{S}\left(\mathbb{R}^M\right)\,. \tag{D.3}$$

In this case Eq. (B.1) takes the form

$$-\sum_{\vec{i}\neq\vec{j},\alpha} \bar{a}_{\vec{i}\alpha\leftarrow\vec{j}\alpha} \left( \frac{d_{\vec{i}\alpha}}{p_{\vec{i}\alpha}} - \frac{d_{\vec{j}\alpha}}{p_{\vec{j}\alpha}} \right)^2 p_{\vec{j}\alpha} + \mathcal{O}(\varepsilon)\,, \quad \text{with} \quad p_{\vec{j}\alpha} = \pi_{j_1} \dots \pi_{j_n} q_\alpha\,. \tag{D.4}$$

Suppose now that $a_{1\leftarrow 2}$ is the only negative rate of $R^{(t)}$. We want to show that despite the onset of non-Markovianity, the Fisher distance continues to decrease for all the points in the image of $\bar{T}^{(t,0)}$, i.e., the derivative expressed in Eq. (D.4) is negative. Then, consider for the moment the positive contributions to Eq. (D.4). These are given by

$$-\sum_{l=1}^{n} a_{1\leftarrow 2} \left( \frac{d_{\vec{j}_{l=1}\alpha}}{p_{\vec{j}_{l=1}\alpha}} - \frac{d_{\vec{j}_{l=2}\alpha}}{p_{\vec{j}_{l=2}\alpha}} \right)^2 p_{\vec{j}_{l=2}\alpha}\,, \tag{D.5}$$

where $\vec{j}_{l=2}$ is the string $\vec{j}_{l=2} = (j_1, \ldots, j_{l-1}, 2, j_{l+1}, \ldots, j_n)$. We can now compare Eq. (D.5) with the following contribution

$$-\sum_{l=1}^{n} a_{2\leftarrow 1} \left( \frac{d_{\vec{j}_{l=1}\alpha}}{p_{\vec{j}_{l=1}\alpha}} - \frac{d_{\vec{j}_{l=2}\alpha}}{p_{\vec{j}_{l=2}\alpha}} \right)^2 p_{\vec{j}_{l=1}\alpha}, \tag{D.6}$$

where we defined $\vec{j}_{l=1}$ in analogy with $\vec{j}_{l=2}$. Summing up Eq. (D.5) and Eq. (D.6) then turns out to be negative whenever $p_{\vec{j}_{l=1}\alpha} a_{2\leftarrow 1} \geq p_{\vec{j}_{l=2}\alpha} |a_{1\leftarrow 2}|$, i.e., $\pi_1 a_{2\leftarrow 1} \geq \pi_2 |a_{1\leftarrow 2}|$, in complete analogy with what happened in the single copy case.

### D.2.1  Theorem 2: quantum case

Theorem 2 holds also in the quantum setting. The extension of the proof makes use again of Lemma 2. In particular, we exploit a quantum dynamical map $\mathcal{T}^{(t,0)}$ with a strong dephasing which reduces the states to be (almost) classical. That is, consider the evolution given

$$\mathcal{T}^{(t,0)}[\rho] = \mathcal{D}^{(\varepsilon_2)} \circ \mathcal{F}_{\pi}^{(\varepsilon_1)}[\rho], \tag{D.7}$$

where

$$\mathcal{F}_{\pi}^{(\varepsilon_1)}[\rho] = (1-\varepsilon_1)\pi + \varepsilon_1\rho, \tag{D.8}$$

$$\mathcal{D}^{(\varepsilon_2)}[\rho] = (1-\varepsilon_2)\rho_D + \varepsilon_2\rho, \quad \rho_D = \sum_i |i\rangle\langle i| \rho |i\rangle\langle i|, \tag{D.9}$$

and $|i\rangle$ is an eigenbasis of $\pi$ (so that $\mathcal{D}$ is the dephasing operator in the basis of $\pi$).

Now thanks to Lemma 2 for small enough $\varepsilon_2$, one can compute the Quantum Fisher Information and its instantaneous variation by reducing it to its classical value. To be precise, as the Theorem considers multiple copies of the channel and ancillary degrees of freedom, one needs to verify that also in this case the evolution of the Fisher Information collapses onto the classical case. Consider any initially prepared $\rho$ and $\rho + \delta\rho$ quantum states of $\mathcal{H}^{\otimes n} \otimes \mathbb{C}^M$, where $\mathcal{H}$ is the $N$-dimensional space of the single-copy channel, and an $M$-dimensional ancilla is allowed. Then the evolution to time $t$ is given by

$$\bar{\mathcal{T}}^{(t,0)} = \mathcal{T}^{(t,0)\otimes n} \otimes \mathbb{1}_M, \tag{D.10}$$

where $\mathcal{T}^{(t,0)}$ is as in Eq. (D.7). By defining $\sigma$ to be the reduced state of $\rho$ on the ancillary degrees of freedom, one has

$$\bar{\mathcal{T}}^{(t,0)}[\rho] = \pi^{\otimes n} \otimes \sigma + \mathcal{O}(\varepsilon_1). \tag{D.11}$$

At the same time, it also holds that

$$\bar{\mathcal{T}}^{(t,0)}[\delta\rho] = \varepsilon_1 \delta\rho_D + \mathcal{O}(\varepsilon_1\varepsilon_2), \tag{D.12}$$

where $\delta\rho_D = \mathcal{D}^{(0)\otimes n} \otimes \mathbb{1}_M[\delta\rho]$. As such it can be expressed as

$$\delta\rho_D = \sum_{\gamma} \theta_D^{(\gamma)} \otimes \omega^{(\gamma)}, \quad [\theta_D^{(\gamma)}, \pi^{\otimes n}] = 0, \tag{D.13}$$

where the $\theta_D^{(\gamma)}$ are operators on $\mathcal{H}^{\otimes n}$ and $\omega^{(\gamma)}$ on $\mathbb{C}^M$. The quantum Fisher Information can then be computed as

$$\frac{\varepsilon_1^2}{2} \text{Tr}\left[ \delta\rho_D \, \mathbb{J}_{f,\pi^{\otimes n}\otimes\sigma}^{-1}[\delta\rho_D] \right] + \mathcal{O}(\varepsilon_1^3) + \mathcal{O}(\varepsilon_1^2\varepsilon_2). \tag{D.14}$$

By substituting the expression for $\delta\rho_D$, we get that the leading order is

$$\frac{{\varepsilon_1}^2}{2}\sum_{\gamma,\gamma'}\text{Tr}\left[\theta_D^{(\gamma)}\mathbb{J}_{f,\pi^{\otimes n}}^{-1}\theta_D^{(\gamma')}\right]\text{Tr}\left[\omega^{(\gamma)}\mathbb{J}_{f,\sigma}^{-1}\omega^{(\gamma')}\right]:=\frac{{\varepsilon_1}^2}{2}\sum_{\gamma,\gamma'}\mathcal{M}_{\gamma\gamma'}^{(1)}\mathcal{M}_{\gamma\gamma'}^{(2)}.\tag{D.15}$$

Given that only the first trace is time-dependent (the evolution on the ancillary degrees of freedom is trivial), the instantaneous derivative of the above equation can be written as

$$\frac{{\varepsilon_1}^2}{2}\sum_{\gamma,\gamma'}\left(\frac{\mathrm{d}}{\mathrm{d}t}\mathcal{M}_{\gamma\gamma'}^{(1)}\right)\mathcal{M}_{\gamma\gamma'}^{(2)},\tag{D.16}$$

that is, as the trace-product of two matrices, $\frac{\mathrm{d}}{\mathrm{d}t}\mathcal{M}^{(1)}$ and $\mathcal{M}^{(2)}$. We now notice that $\mathcal{M}^{(2)}$ is positive definite, therefore it is enough for $\frac{\mathrm{d}}{\mathrm{d}t}\mathcal{M}^{(1)}$ to be negative definite in order for the product to be $\leq 0$.

To prove the quantum version of Theorem 2 it is then sufficient to provide a non-Markovian evolution for which $\frac{\mathrm{d}}{\mathrm{d}t}\mathcal{M}^{(1)}$ is negative definite. It is also enough to consider the case in which $\mathcal{M}^{(1)}$ reduces to its classical value, due to $[\theta_D^{(\gamma)},\pi]=0$. That is

$$\mathcal{M}_{\gamma\gamma'}^{(1)}=2\left\langle\boldsymbol{\theta}^{(\gamma)},\boldsymbol{\theta}^{(\gamma')}\right\rangle_\pi,\tag{D.17}$$

where $\boldsymbol{\theta}^{(\gamma)}$ and $\pi$ are the vectors given by the diagonal components of $\theta^{(\gamma)}$ and $\pi$ respectively. It is then clear that choosing at time $t$ a dynamics of the form

$$\mathcal{T}^{(t+\delta t,t)}=\mathbb{1}+\delta t\mathcal{L},\tag{D.18}$$

where $\mathcal{L}$ is a semi-classical Lindbladian with rates $a_{i\leftarrow j}$ and jump operators $|i\rangle\langle j|$

$$\mathcal{L}[\rho]=\sum_{i\neq j}a_{i\leftarrow j}\left(|i\rangle\langle j|\rho\,|j\rangle\langle i|-\frac{1}{2}\{|j\rangle\langle j|,\rho\}\right),\tag{D.19}$$

will induce the classical dynamics locally described by the rates $a_{i\leftarrow j}$ on the diagonal subspace of quantum states.

Finally, the classical version of Theorem 2, given above D.2, ensures that there are instances in which at least one of the $a_{i\leftarrow j}$ is negative (and hence $\mathcal{T}^{t+\delta t,t}$ non-CP) while $\frac{\mathrm{d}}{\mathrm{d}t}\mathcal{M}^{(1)}$ is negative definite. This concludes the proof.

## D.3 Proof of Theorem 3

Despite the negative result given by Theorem 2, we provide here an explicit construction to convert a non-Markovianity witness based on the trace-distance to one that uses the Fisher metric by exploiting a particular kind of post-processing.

Consider, in fact, the case in which during a non-Markovian dynamics there is some $\delta\rho$ such that $\frac{\mathrm{d}}{\mathrm{d}t}|\delta\rho|_{\text{Tr}}^2$ is increasing in time. As we saw in Appendix A, to guarantee the existence of such $\delta\rho$, it is sufficient to use an ancilla, namely by considering the dynamics $T^{(t,0)}\otimes\mathbb{1}_2$ for the classical case, and $\mathcal{T}^{(t,0)}\otimes\mathbb{1}_{N+1}$ for the quantum case.

Thanks to Lemma 3, we also know that on the base-point $\rho_{\delta\rho}$ it holds that

$$\frac{\mathrm{d}}{\mathrm{d}t}D_{\text{Fish}}\left(\rho_{\delta\rho},\rho_{\delta\rho}+\delta\rho\right)=\frac{1}{2}\frac{\mathrm{d}}{\mathrm{d}t}|\delta\rho|_{\text{Tr}}^2>0.\tag{D.20}$$

It is useful to repeat the computation of this derivative here. We obtain

$$\frac{\mathrm{d}}{\mathrm{d}t}D_{\text{Fish}}\left(\rho_{\delta\rho},\rho_{\delta\rho}+\delta\rho\right)=\frac{1}{2}\frac{\mathrm{d}}{\mathrm{d}t}\sum_i\frac{\delta\rho_i^2}{(\rho_{\delta\rho})_i}=\sum_i\left(\frac{\delta\rho_i\,\delta\dot{\rho}_i}{(\rho_{\delta\rho})_i}-\frac{1}{2}\frac{\delta\rho_i^2\,(\dot{\rho}_{\delta\rho})_i}{(\rho_{\delta\rho})_i^2}\right).\tag{D.21}$$

The last term does not contribute to the sum: in fact, the first term already gives $\frac{1}{2}\frac{d}{dt}|\delta\rho|^2_{Tr}$, so the last term must be zero due to Lemma 3. On the other hand, it is not difficult to carry out the explicit calculation, giving

$$\sum_i \frac{\delta\rho_i^2(\dot\rho_{\delta\rho})_i}{(\rho_{\delta\rho})_i^2} = \left(\sum_i \frac{\delta\rho_i^2(\dot\rho_{\delta\rho})_i}{\delta\rho_i^2}\right)\left(\sum_j |\delta\rho_j|\right)^2 = 0, \tag{D.22}$$

where we used the expression of $\rho_{\delta\rho}$, together with the fact that $\dot\rho_{\delta\rho}$ is traceless. Hence, we showed that despite the explicit dependence of the Fisher information on $\rho_{\delta\rho}$, there is no contribution in its derivative coming from $\dot\rho_{\delta\rho}$. In some sense, we can deduce that the base-point could be *frozen* and the Fisher information would still have a backflow.

This intuition inspires the following construction. Define the stochastic operator $F$ as

$$F_\pi^{(\varepsilon)}[\sigma] = (1-\varepsilon)\pi + \varepsilon\sigma, \tag{D.23}$$

where $\pi$ is some arbitrary state and $\varepsilon$ is a small parameter. That is, the operator $F$ is an almost-complete erasure of prior information, sending all the states close to $\pi$. If $\sigma$ and $\tau$ are two states, their difference $\Delta := \sigma - \tau$ is transformed as:

$$F_\pi^{(\varepsilon)}[\Delta] = F_\pi^{(\varepsilon)}[\sigma] - F_\pi^{(\varepsilon)}[\tau] = \varepsilon\Delta. \tag{D.24}$$

In particular, the Fisher information between $p$ and $p + d$ transforms under the application of this filter as

$$D_{Fish}^2\left(F_\pi^{(\varepsilon)}[p], F_\pi^{(\varepsilon)}[p+d]\right) = \sum_i \varepsilon^2\frac{d_i^2}{\pi_i} + \mathcal{O}(\varepsilon^3) \simeq D_{Fish}^2(\pi, \pi+\varepsilon d) + \mathcal{O}(\varepsilon^3). \tag{D.25}$$

Then, choose $\pi$ as $\pi := \rho_{\delta\rho}$. Then, we can construct a witness of the non-Markovianity of the evolution at time $t$ through the Fisher information, by applying the filter $F_{\rho_{\delta\rho}}^{(\varepsilon)}$ to the final state. This gives:

$$\frac{d}{dt}D_{Fish}^2\left(F_{\rho_{\delta\rho}}^{(\varepsilon)}[\rho(t)], F_{\rho_{\delta\rho}}^{(\varepsilon)}[\rho(t)+\delta\rho(t)]\right) \tag{D.26}$$

$$= \lim_{dt\to 0}\left(\frac{D_{Fish}^2\left(F_{\rho_{\delta\rho}}^{(\varepsilon)}[\rho(t+dt)], F_{\rho_{\delta\rho}}^{(\varepsilon)}[\rho(t+dt)+\delta\rho(t+dt)]\right)}{dt}\right.$$

$$\left. -\frac{D_{Fish}^2\left(F_{\rho_{\delta\rho}}^{(\varepsilon)}[\rho(t)], F_{\rho_{\delta\rho}}^{(\varepsilon)}[\rho(t)+\delta\rho(t)]\right)}{dt}\right) \tag{D.27}$$

$$= \lim_{dt\to 0}\left(\frac{D_{Fish}^2(\rho_{\delta\rho}, \rho_{\delta\rho}+\varepsilon\,\delta\rho(t+dt)) - D_{Fish}^2(\rho_{\delta\rho}, \rho_{\delta\rho}+\varepsilon\,\delta\rho(t)) + \mathcal{O}(\varepsilon^3)}{dt}\right) \tag{D.28}$$

$$= \varepsilon^2\frac{d}{dt}|\delta\rho(t)|^2_{Tr} + \mathcal{O}(\varepsilon^3), \tag{D.29}$$

where in Eq. (D.28) we used the result from Eq. (D.25). Regarding the last equality, on the other hand, it should be noticed that in Eq. (D.28) $\rho_{\delta\rho}$ does not depend on time. Still, thanks to the remarks made above, we know that $\dot\rho_{\delta\rho}$ does not contribute to the derivative, so we can apply Lemma 3. Incidentally, in the main text we use the notation $F_d := F_{\rho_{\delta\rho}}^{(\varepsilon)}$, where $d$ is the diagonal part of $\delta\rho$.

Now, the quantity constructed is contractive under Markovian dynamics, as the trace distance is contractive. On the other hand, by choosing $\delta\rho$ to be a witness of non-Markovianity

for the trace distance, we also obtain a witness in this scenario. There is a shortcoming to this construction, though: in the definition of the post-processing $F_{\rho_{\delta\rho}}^{(\varepsilon)}$ one uses the perturbation $\delta\rho(t)$, and not just $\delta\rho(0)$ (in fact, to be more precise, the filter is actually given by $F_{\rho_{\delta\rho(t)}}^{(\varepsilon)}$). In this way, one has to know in advance the structure of the dynamics to provide an explicit construction. Still, this example serves more as a proof of principle of the possibility of designing post-processing filters to exploit the Fisher information for the detection of non-Markovianity.

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
