# Peer review of "Characterizing (non-)Markovianity through Fisher Information"

_SciPost Physics, doi:SciPost Phys. 15, 014 (2023)_

## Round 4 · Referee Report · Anonymous · 2023-1-17

Strengths

1) The new version of this article is easier to follow, and welcome details have been introduced in the introduction.
2) The unclear points have been clarified.
3) The example, as requested by the first referee, shows the relevance of the results in a way far more efficient than other parts of the main text.

Weaknesses

1) One or two sentences of connection between the case study and a real physical model could be interested. There is no link between the example and previous studies of non-markovian dynamics

Report

This new version could be published in its current form, but nonetheless, a few words of connection between the case study and the real world can be helpful. Is it related to a real physical system, if yes, which one? Also, is this model already studied in the literature (new reference ?) ?

This kind of clarification could be interesting for readers who want to apply the results of this paper to physical systems, but it does not impact the relevance or the quality of the results.

Requested changes

If applicable, a few words of connection between the case study and a physical system.

---

## Round 4 · Referee Report · Anonymous · 2023-1-25

Report

Since all the required changes have been made, I support the publication of this paper.

---

## Round 4 · Author Response

RESPONSE TO REFEREE 1

We thank the Referee for their feedback on our work.

<< The description and the characterization of the dynamics of open quantum systems is a subject of fundamental interest and a basic prerequisite for applications in quantum computing and more generally in quantum technologies. The goal of this paper is a key objective of this field. In the past few years, many papers have proposed different measures and ways to characterize the non-Markovian character of an open quantum system. The results of this paper are interesting in this direction. >> We thank the Referee and agree with the assessment of importance and originality of our work. The Referee points out two main criticisms: one regarding the clarity and technicality of our results

<< However, I think that this paper cannot be published in its current version in SciPost Physics and should be submitted to a more mathematical journal. The derivation of the different results is quite technical and difficult to follow for a non-expert. >> We stress that, in the main text of the paper, we only use elementary algebra and analysis on finite-dimensional real vector spaces. Part of the technicality that is present in the paper is due to the need of rigorously formalizing physical statements about the contractivity of the Fisher metric in relation to Markovian evolutions. Moreover, we explain the intuitive meaning of the theorems presented, whose proofs are presented in a reader-friendly form, moving technical details to the appendix. In the new version we also improved the clarity of some discussions, as well as increasing the background on Fisher Information metric.

<< Moreover, much of the paper focuses on classical dynamics whereas (as mentioned above) the role and characterization of non-Markovianity has been mainly studied in quantum physics. This aspect is only briefly described in the conclusion of the paper. >> The choice of presenting in the main text the discussion of classical dynamics only is due to two main reasons: 1. All the physical intuition of our results can be presented already for classical system. The quantum scenario does not need new physical insights to be understood, rather it only requires technical lemmas that are used to reduce the proofs to classical dynamics. The advantage of presenting mainly the discussion of the classical case is then clear, as it is easier to understand physically and mathematically. 2. The lift of the results to the quantum scenario involves mathematical tools that are more advanced, known to a smaller fraction of the public, is technically tedious, and results in physical statements that are identical, mutatis mutandis, to the same physical statements in the case of classical dynamics. In the new version of the manuscript we stress this point not only at the end of the Framework Section (Sec. 2) and in the Conclusions (Sec 5.1), but also in the introduction, improving the readability for readers from the quantum information community and others.

<< In addition, the different theoretical results are not applied to standard physical examples in order to show their importance and their relevance. >> In the new version of the manuscript we added paragraph (“A case study”, at the end of Sec.3), in which we showcase the relevance of our proposed analysis (based on the behaviour of the Fisher metric) by applying it to a vast class of dynamics whose non-Markovianity cannot be witnessed by the standard Trace-distance based approach (without ancillas).

RESPONSE TO REFEREE 2

<< The idea of the paper is interesting. Linking the Fisher information to the study of Markovianity may be very useful to connect several ideas in quantum information processing, where the Fisher information has been widely used in recent studies of open quantum systems. >> We thank the Referee for their overall positive assessment, the attentive analysis and useful feedback on our work. In the following, we clarify all the points raised by the referee.

<< 1) Efforts have been made to present state of the art results of Markovian processes (this is welcome) but the presentation of the Fisher distance is reduced to almost nothing. Since it is a key notion of the paper, I think that it must be presented with more details. I think that it will be useful for most of the readers, which may be more knowledgeable on Markovian processes than Fisher's distance. We agree with the referee and in the new version we increased the background dedicated to the Fisher Information. 2) The authors use extensively the approximation of D_Fish (defined in Eq. 2), which gives us a local information on the space of probability measure, in the vicinity of a measure p. This is not a problem for most of the results, but this is very dangerous when it is extrapolated to the global space. […] The conditions where such a 'change of scale' can be made must be clearly identified, specifically in Thm. 1. >> Although we make heavy use of the local approximation of the Fisher Distance, this is enough to formally prove all the results at a global level. In practice, all of our proofs are based on statements regarding two situations: situation 1) the evolution is stochastic-divisible, and the Fisher Distance contracts locally and globally. Notice that the local contraction of the metric, is enough to guarantee the global contractivity, thanks to the triangular inequality (specifically thanks to the fact that the original geodesics curve between two points will contract its length, which gives an upperbound to the new geodesics length). We point out this fact in the main text in the new version. Besides, the contractivity of the Fisher distance under stochastic maps (both locally and globally) is known already in the literature. situation 2) The evolution is not stochastic divisible, then there exists two points which are instantaneously increasing their distance. For this kind of statements, one only needs a single instance of such points, therefore such instance can be found locally. All the presented results of Theorems 1-2-3 belong to this class of statements, for according to which everything we present is mathematically correct, even without using the global expression of D_Fish. Theorem 4 is instead valid only locally, and has the physical meaning of being valid for “small errors” between prior information and actual initial state. We specify this in the text. In the new version, we clarify all these issues by adding key comments throughout the manuscript.

<< 2) In thm 1 (or before) it may be interesting to specify that this is not a "true" non-Markovian process which is studied, but the linearization of a process whose domain of definition is extended to the full space, and not a small set around the point used for the linearization. This may simplify the comment on the points that "cannot be achieved physically" (page 6). >> We thank the Referee for the feedback. In the new version, we in fact address this point just before Theorem 1, clarifying what is the object we study and its mathematical interpretation.

<< 3) I am not convinced by the comparison between Fisher distance and trace-distance. If we take the following equation as the definition of DFish […]. So, the trace distance is a distance adapted to measure points in the simplex, and the Fisher distance is adapted to measure distances in the sphere. Since there is a diffeomorphism between the two spaces, I expect that […]. If a dynamical map contracts D_Tr it must contract D_Fish and reciprocally. >> The diffeomorphism between two distances preserves certain properties, such as the definition of open sets and the continuity of functions. However, it does not preserve the geodesics length, nor the geodesics trajectory, nor the rate of variation of geodesic length. For example, it is possible to translate two vector p and q, to p+d and q+d . In such case, the Trace distance is preserved but not the Fisher distance. This means that the rate of change of the former is zero, while the latter can be positive, zero, or negative. If both points p and q vary in time arbitrarily, the rate of change of D_Tr and D_Fish do not have the same sign in general.

<< 6) Following the point 5), I'm not convinced by the fact that "the Fisher information metric [...] natural object to study". But I agree that it gives us an interesting point of view that may simplify some calculations. >> We agree with the Referee, and in the new abstract we toned-down to “the Fisher information metric emerges as a natural object to study”

<<4) The derivation of Eq. 21 must be clarified. It is not clear to me if the level of approximation is consistent with the one of Eq. 2, and hence, if a hidden mistake may be hidden in Thm. 4.>> In the new version the same equation has become Eq.(29), and the text leading to it has been modified to improve clarity.

Requested changes << 1) Clarify the working hypothesis of Thm.1, according to the previous comments. 2) Don't the use of the approximation of D_Fish when it can be easily avoided. The approximation is not always helpful, and it can lead to false conclusions.>> We clarified these issues and modified the main text according to the discussion presented above.

<< 3) Typo in Eq. 14 ?>> In the new version this is Eq.(18), there is no typo.

<<4) (optional) Remove or specify the sections in the last sentence of Appendix B.>> We rephrased the sentence.

---

## Round 4 · List of Changes

- Above Eq.(1), sentences added from "Notice that..." to " ...(cf. Appendices C and D)."
- Above Eq.(2), specificed "at leading order O(|d|^2)"
- All the discussion around Eq.s (3)-(4) has been modified and increased with more details, including footnote 1, in order to give additional background on the Fisher distance.
- Footnote 1 contains now a small technical discussion on the finite Fisher distance.
- Above Eq.(17) an improved explanation is given for the meaning of the rate of contraction outside the image of the map, as suggested by Referee 2.
- The proof of Theorem 1 has been modified to improve clarity.
- Footnote 4 has been added to improve clarity.
- After Theorem 3, a whole paragraph ("A case study") has been added to provide a specific example of non-Markovianity that can be witnessed by the Fisher distance, but not by the Trace distance.
- Modifications above Eq.(29) to improve clarity.
- Additional discussion below Eq.(34) to improve clarity.
- Small modifications to sentences in the conclusion and in the appendix.
- Minor corrections and revision

---

## Round 5 · Author Response

Dear Editor and Referees,
Thank you for assessing our work.
After the latest round of reports, we hereby resubmit the final version of the paper with minor revision.
As suggested by Referee 1, we included a comment - footnote 5, below Eq.(27) - to connect the specific example we use in the main text (to showcase some of our results) to known amplitude damping channels with thermal asymptotic states.

Best regards,
The Authors.

---

## Editorial Decision

published